# Artificial Generals Intelligence:
# Mastering Generals.io with Reinforcement Learning

**Matej Straka**[1], **Martin Schmid**[1, 2]

`strakammm@gmail.com, schmidm@kam.mff.cuni.cz`

[1]**Charles University, Prague, Czech Republic**
[2]**EquiLibre Technologies, Inc.**

## Abstract

We introduce a real-time strategy game environment built on Generals.io, a game that hosts thousands of active players each week across multiple game formats. Our environment is fully compatible with Gymnasium and PettingZoo, capable of running thousands of frames per second on commodity hardware. Our reference agent—trained with supervised pre-training and self-play—hits the top 0.003% of the 1v1 human leaderboard after just 36 hours on a single H100 GPU. To accelerate learning, we incorporate potential-based reward shaping and memory features. Our contributions—a modular RTS benchmark and a competitive, state-of-the-art baseline agent—provide an accessible yet challenging platform for advancing multi-agent reinforcement learning research.

## 1 Introduction

Games have played a pivotal role in the development of modern reinforcement learning (RL), acting as crucial benchmarks to advance the field of artificial intelligence. RL has been applied across various game types, starting with early successes in single-agent control like Atari (Mnih et al., 2013), and leading to significant milestones in perfect-information adversarial board games such as Go and Chess through self-play, exemplified by AlphaZero (Silver et al., 2017). Alongside these, RL has made strides in imperfect-information games such as poker (Moravčík et al., 2017; Brown & Sandholm, 2018) and Stratego (Perolat et al., 2022), which require reasoning under uncertainty and strategic deception. A distinct category emerged with games like Diplomacy (Meta Fundamental AI Research Diplomacy Team, 2022), where RL has been utilized in environments where strategic alliances and verbal communication are key game mechanics, leveraging large language models (LLMs) to facilitate this communication. Another major venue for RL research has been large-scale, real-time multi-agent video games under partial observability, as evidenced by powerful systems developed for StarCraft II (Vinyals et al., 2019) and Dota II (Berner et al., 2019). These diverse game environments collectively pose significant challenges—from low-level control and long-horizon planning under incomplete information to intricate multi-agent coordination—driving crucial advances in reinforcement learning and game theory.

We present a real-time strategy environment based on the browser game Generals.io[1]. Our environment aims to provide an interesting benchmark with challenges comparable to those in large strategic video games such as StarCraft II, while being lightweight enough to allow RL experimentation on modest hardware. Moreover, Generals.io maintains an active community of several thousand weekly participants across multiple game formats. Each format hosts a quarterly championship tournament for top-ranked players, fostering a robust and dynamic competitive ecosystem. Although bots are

---

[1]`https://generals.io`

allowed to compete, even the most advanced have yet to attain a top placement in these tournaments, underscoring the challenge and potential of this platform as a benchmark for agent development.

There are two main contributions of this paper: first, we introduce a real-time strategy environment that is vectorized, compatible with Gymnasium (Towers et al., 2024) and PettingZoo (Terry et al., 2021), and capable of running thousands of frames per second on commodity hardware; second, we develop a PPO-based agent that achieves top-tier performance on the 1v1 human leaderboard after 36 hours of training on a single H100 GPU.

The remainder of the paper is structured as follows. In Section 2, we situate our work within existing MARL benchmarks and prior Generals.io work. Section 3 formalizes the game mechanics and the partial observability model. In Section 4, we introduce our environment. Section 5 then describes our reference agent—covering behavior cloning, self-play, reward shaping, feature design, and network architecture. Section 6 evaluates our agent against humans and existing bots. Finally, Section 7 summarizes our contributions and outlines the directions for future research.

## 2 Related Work

Multi-agent RL environments can be broadly divided into cooperative, fully competitive, and mixed settings. In cooperative scenarios such as the StarCraft Multi-Agent Challenge (SMAC), agents rely on local observations to collaboratively micromanage unit control (Samvelyan et al., 2019); fully competitive environments like FightLadder (Li et al., 2024) involve direct competition between agents; and mixed settings, exemplified by hide-and-seek (Baker et al., 2020), feature teams that cooperate internally while competing against opponents to achieve their objectives.

GENERALS can be played in both fully competitive (1v1, free-for-all) and mixed (e.g., 2v2) modes. Bhatia et al. (2023) introduce a Redis-based framework for data collection and bot integration with Generals.io; however, their choice of tools is ill-suited for machine learning applications, due to their lack of scalability. They also present a rule–based agent, `Flobot`, which was once competitive but is now substantially outperformed by more recent approaches.

Later, Xu et al. (2018) highlighted GENERALS as an efficient and cost-effective research platform that offers challenges comparable to Dota 2 and StarCraft II, while supporting rapid simulation and a large online player base. They propose a Hierarchical Agent with Self-Play (`HASP`) and report a 77% win rate against `Flobot`.

Prior to this work, the community-developed agent `Human.exe`[2], created by hobbyist `EklipZ`, was considered state-of-the-art, consistently ranking among the top players on the Generals.io leaderboards. It offers a formidable challenge even to expert human opponents and obtains high winrates against both `Flobot` and `HASP`. Remarkably, `Human.exe` was engineered without any machine learning techniques, relying solely on heuristics and classical algorithms based on deep domain expertise. Moreover, `Human.exe` is capable of playing 1v1, free-for-all, and 2v2 formats.

## 3 Game Description

GENERALS is played on an $H \times W$ grid. Each player seeks to be the last to stand by capturing the opponent's *general*. This is achieved by expanding territory, managing resources, and maneuvering armies to breach enemy defenses. In this section, we provide a detailed overview of the game's mechanics.

**Grid Generation.** At the start of each match, an $H \times W$ grid is populated with four cell types: *plain* (traversable), *mountain* (impassable), *general* (player base cell), and *castle* (army generating cell). The grid generator reproduces terrain statistics from the web-based Generals.io implementation, populating the grid with approximately 20% mountain cells and 9–11 castles (each randomly

---

[2] https://github.com/EklipZgit/generals-bot

initialized with 40–50 neutral units). Player generals are placed at least 15 breadth-first-search steps apart, and each player is guaranteed at least one castle within a BFS radius of 6. An example of a grid layout can be seen in Figure 1.

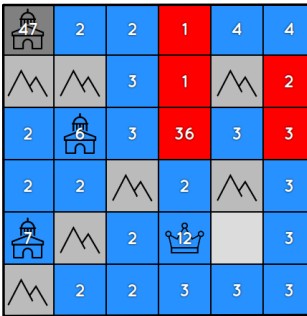 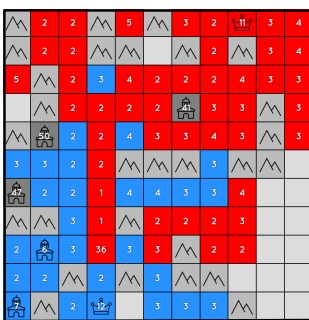 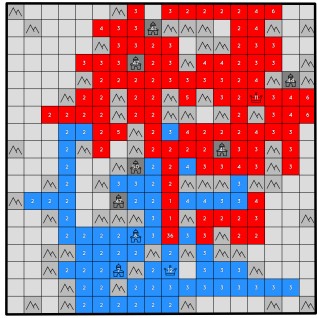

Figure 1: Three zoom levels of the same game state are shown. Cells of different colors belong to different players. Each player has one base, depicted as a crown. Numbers in the cells represent the number of units in that cell. A player can move units to any neighboring cell, except in the direction of mountains.

**Ownership and Partial Observability.**   Cells may be owned by a player or remain neutral.  A fog-of-war mechanic restricts the view of each player to (i) all cells they own and (ii) the immediate eight neighbor ("Moore") neighborhood around those cells; all other cells are hidden (see Figure 2). In parallel, a global scoreboard displays each player's total owned cells and the aggregate army count, providing global information on player statistics.

**Army Growth.**   Players expand their armies through territorial and base generation mechanisms. Every 25 turns, each player gains one unit on each grid cell they control, so having more territory accelerates army growth. In addition, each player's base generates one unit per turn. Neutral castles can be captured at the upfront cost of 40–50 units; once captured, they too generate one unit per turn, improving long-term production.

**Movement and Combat.**   Gameplay advances in two half-turns per turn, with all players issuing moves simultaneously (each half-turn lasts 500ms online). Although this interval seems brief, players routinely premove, making the allotted time more than sufficient for decision-making.  A move consists of selecting a source cell and a direction, dispatching either all but one unit or exactly half of the units (player's choice) from the source to the adjacent cell. When player's units enter an enemy-occupied cell, combat ensues: army strengths are subtracted, and the larger force prevails, occupying the cell with surviving army. In the event of equal forces, ownership remains with whichever player occupied the cell first. Although moves appear to occur simultaneously, in a few edge cases they are resolved in a specific order, which we omit here for brevity.

**Strategic Complexity.**   Several interacting factors deepen the strategic challenge. Fog-of-war injects uncertainty, forcing players to deduce the distribution of the enemy army and the location of the enemy base from partial observations and scoreboard fluctuations. Deception emerges naturally: feints and diversions can bluff opponents about true intentions and base positions. Tempo is critical: capturing territory just before the 25-turn reinforcement yields a disproportionate advantage, while overexpansion can leave armies too spread out for decisive attacks. Capturing castles requires an upfront army investment that may temporarily expose vulnerabilities despite future production benefits. The interplay of uncertainty, deception, information gleaned from global metrics, tempo management, and resource allocation creates a rich strategic landscape in which small decisions can cascade into game-deciding outcomes.

In the left part of the Figure 1, the red player can capture the blue base in two moves by moving his 36 units toward the blue's base. We also see that the blue player invested in two castles, whereas the red player focused on territorial expansion. Strategically, the blue player appears to have overcommitted his resources to capturing castles, allowing the red player to exploit this imbalance.

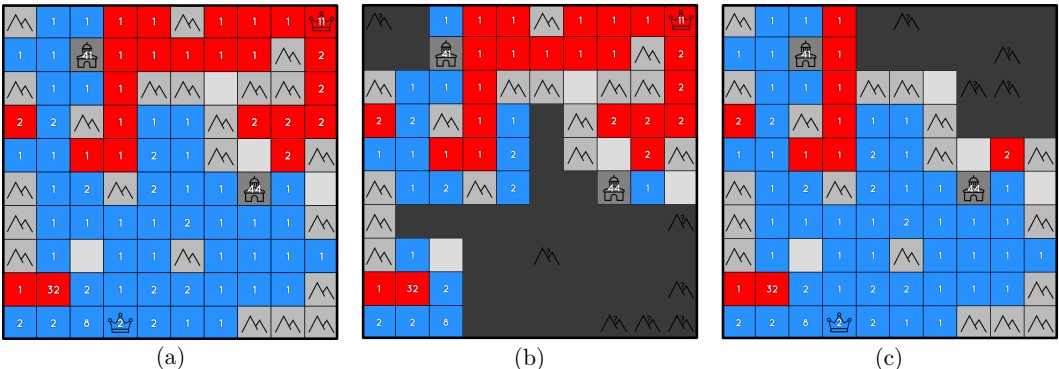

Figure 2: Three views of the same game state: (a) perfect information view; (b) point of view of the red player; and (3) point of view of the blue player.

## 4    Environment

Our implementation offers a *vectorized* and customizable testbed for GENERALS. It exposes both the Gymnasium and PettingZoo interfaces, provides built-in support for recording and replaying matches, and integrates a PyGame renderer for visual inspection. In addition to the default grid generation procedure, the generation may be performed manually via a text file or procedurally by defining high-level parameters (e.g. grid dimensions, mountain density, or castle count). Trained policies can be deployed directly to official Generals.io servers, allowing rigorous comparison with human and third-party agents. Benchmarks on a 12-core CPU with 12 parallel environments yield on the order of **3,500** frames per second. The complete source code, usage guide, and exemplar scripts for training and deployment are available on GitHub[3].

**Observation Space.**    We provide two equivalent representations of the state of the game. In the `dict` format, each entry is either a matrix (e.g., binary matrix indicating mountains) or a scalar statistic (e.g., `opponent_land_count`). In the `tensor` format, these matrices and scalars are lifted into separate feature planes and concatenated into a 3D tensor $\mathbb{R}^{C \times H \times W}$, directly compatible with convolutional architectures.

**Action Space.**    Agent actions are represented by a five-element vector:

$$[\texttt{pass}, \texttt{i}, \texttt{j}, \texttt{direction}, \texttt{split}].$$

The binary flag `pass` determines whether the agent abstains from movement on the current step. The coordinates `i` and `j` identify the origin cell of the intended movement. The value of `direction` specifies one of four cardinal directions, and the binary `split` indicates whether to move all units (0) or only half of them (1).

**Rewards.**    By default, we employ a sparse reward, assigning a reward of +1 to the winner and -1 to the loser. In addition, we provide a flexible interface that allows for easy customization of reward functions. Furthermore, we offer a set of pre-implemented utilities that compute various game-related statistics, which can be leveraged to design more sophisticated reward mechanisms.

---

[3]https://github.com/strakam/generals-bots

# 5 Agent

Our agent development consists of two main steps. First, *behavior cloning* on curated expert replays establishes an initial policy. This policy is then refined via *self-play fine-tuning* with reinforcement learning, where the agent iteratively improves by competing against a dynamic pool of its prior versions. To navigate the sparse reward landscape during this reinforcement learning phase, *potential-based reward shaping* is employed to provide more consistent learning signals and guide towards more robust policies. The specifics of these primary training stages and supporting mechanisms are detailed in the following text.

**Behavior Cloning.** A corpus of 347 000 raw replays was obtained from the Generals.io AWS bucket. This dataset has many outliers, for example players willingly not finishing their game, being idle, or being played on different (but similar) versions of the game. We observe that long games are strongly associated with atypical, very often suboptimal behavior of players. Therefore, we filter games longer than 500 turns (equal to 1000 moves). We further resort to games played on the newest patch of the game. There is a trade-off between the size of the dataset and the quality of the dataset. We observe that in our particular scenario, agents perform better when trained on high-quality replays of high-rated players (compared to a larger dataset that includes lower-quality games). Therefore, we further restrict ourselves to games that feature at least one participant with a $\geq$ 70-star rating. This yields a dataset of 16 320 games. We then perform *behavior cloning* by predicting the next move of human players for 3 hours wall clock time on a single `H100` GPU. This behavior cloning agent is already powerful enough to beat mediocre players, but is very narrow in its behavior and can be easily exploited.

**Self-Play Fine-Tuning.** After behavior cloning, we refine the agent via self-play using Proximal Policy Optimization (Schulman et al., 2017). Because episodes span hundreds of steps and our hardware limits batch sizes, we rely on Generalized Advantage Estimation (Schulman et al., 2018) with $\lambda = 0.95$ for stable learning. Each new agent faces its $N$ most recent predecessors, which act via $\arg\max$; we observed that training against stochastic opponents makes new agents focus on exploiting mistakes caused by stochasticity, instead of focusing on more robust gameplay. When a trained candidate achieves a 45 % win-rate versus the current pool (approximately 55 % with both sides using $\arg\max$), it replaces the oldest model. Our top model was trained for 36 hours on a single `H100` GPU with an opponent-pool size of $N = 3$.

**Memory Augmentation.** To play effectively under partial observability, an agent must carry forward critical information from earlier in the game. We therefore augment each raw observation with a small "memory stack" of additional feature-maps that encode what the agent has already discovered. Specifically, before feeding data into our network, we augment observations by information that contains (1) the positions of any castles or generals that have been revealed, (2) which grid cells the agent itself has already explored (so it does not waste time re-searching the same ground), (3) which cells the agent *knows* its opponent has seen (which both signals where the opponent has scouted and indirectly marks territory behind the fog-of-war that the enemy owns), and (4) the last seven moves taken by each side.

However, this hand-crafted memory augmentation does not capture all the details that an agent should remember. For example, the agent does not know *when* it uncovered certain parts of the map, or what were the exact army counts of these cells at that time. We anticipate that incorporating a recurrent architecture or transformer-style memory would allow the network to aggregate and remember those counts and timings, as well as understand that this information is not completely reliable. We leave that extension to future work.

**Architecture.** Our policy is parametrized by a convolutional neural network. We use the same architecture as Perolat et al. (2022), where an U-Net torso processes the augmented observation to produce a board game embedding. This representation is then provided to the value and policy heads

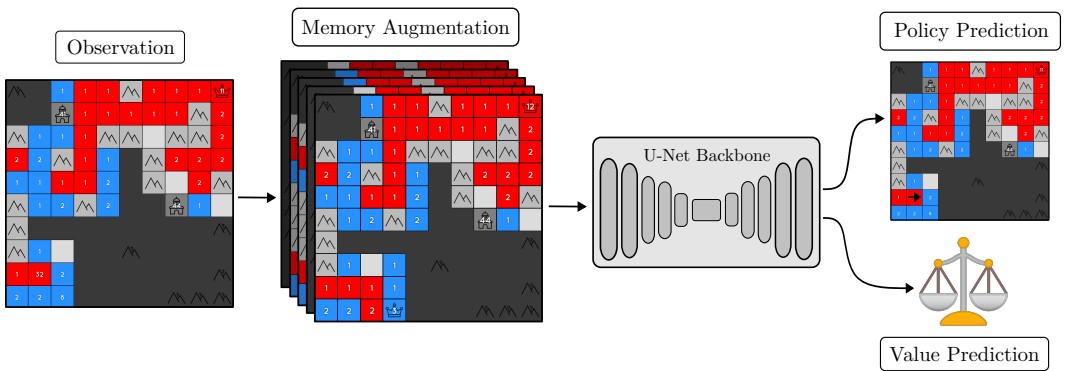

Figure 3: The environment produces game observation encoded as a 3D tensor. This observation is further enriched by memory features and fed into a U-Net feature extractor. The resulting game representation is then passed to value and policy heads, predicting value and action distributions.

(see Figure 3). The policy head returns an $H \times W \times 9$ tensor that encodes a distribution over actions. The nine numbers encode whether the agent wants to `pass` or move `all` or `half` of his units in one of the four directions for each cell.

**Reward Shaping.** Rewards in games are often very sparse, usually just 1 (win) and -1 (lose). Due to long episodes and computational constraints, we provide an additional training signal through potential-based reward shaping (Ng et al., 1999). This type of shaping uses a so-called potential function on states $\phi : S \to \mathbb{R}$ and the shaped reward for performing an action $a$ and moving from state $s \in S$ to $s' \in S$ is expressed as

$$r_{\text{shaped}}(s, a, s') = r_{\text{original}}(s, a, s') + \gamma\phi(s') - \phi(s),$$

where $r_{\text{original}}$ is a simple win/lose reward and $\gamma \in (0, 1]$ is the discount factor. This type of shaping has a property that the set of optimal policies does not change when moving from $r_{\text{original}}$ to $r_{\text{shaped}}$. We define the potential function $\phi$ as the following weighted sum of three log-ratio features:

$$\phi(s) = 0.3\,\phi_{\text{land}}(s) \ + \ 0.3\,\phi_{\text{army}}(s) \ + \ 0.4\,\phi_{\text{castle}}(s),$$

where each sub-potential is

$$\phi_x(s) = \frac{\log\big(x_{\text{agent}}(s)/x_{\text{enemy}}(s)\big)}{\log(\text{max\_ratio})} \quad \text{for } x \in \{\text{land, army, castle}\}.$$

Here max_ratio is the normalization constant used to bound each log-ratio in $[-1, 1]$. The logarithm makes the reward symmetric around a ratio of 1. Consequently, $\phi(s)$ is larger whenever our land, army, or castle counts exceed the opponent's, guiding the agent to states of material advantage. We observe that without reward shaping, the agent quickly converges to an aggressive behavior, where most of the time it focuses entirely on gathering army and attacking, which can be fended off by experienced players who defend well. With reward shaping, the agent's focus shifts toward building material advantage and avoiding unnecessary risks, making it much more robust in all stages of the game.

## 6 Evaluation

We provide an evaluation of our agent, assessing its performance against human experts and other notable bots, including the prior state-of-the-art `Human.exe`. The analysis also covers the impact of key training components and emergent strategic behaviors. Our final agent played 4,700 games

against human opponents under the nicknames `Average Joe` and `zero v3`, and it *consistently ranked within the top 0.003% on the human 1v1 leaderboard, placing it among the top 25 players.* Although it is able to occasionally defeat players in the top 5, such victories are infrequent, with a win rate of approximately 10%. This fact indicates room for further improvement.

In a head-to-head evaluation against `Human.exe`, our agent achieved a 54.82% win-rate across 529 games, with a 95% Wilson score confidence interval of 50.56%–59.01%. In Table 1 we construct an Elo system among bots based on the measured win-rates. Our best performing bot (`zero v3`) obtains a 96% win-rate against `Flobot` which is rooted at 1500 elo. Xu et al. (2018) report a 77% win-rate for `HASP` vs. `Flobot`—though their training directly incorporates `Flobot`, giving them an advantage.

Table 1: Elo ratings from pairwise win-rates between existing agents. Flobot is fixed at 1500.

| Agent | Elo Rating |
|---|---|
| zero v3 (ours) | 2052 |
| Human.exe | 2018 |
| Behavior Cloning (ours) | 1874 |
| HASP | 1710 |
| Flobot | 1500 |

Table 2 summarizes the impact of successive enhancements to the behavior cloning agent. Each step yields an improvement over the previous version, with self-play delivering the largest gain when evaluated against human players. However, pure self-play converged to an overly aggressive, exploitable strategy. To mitigate this, potential-based reward shaping promotes more resourceful gameplay, resulting in a 71.9% win rate against a naive self-play agent. Finally, population-based training further increases robustness, allowing the agent to defeat a variety of opponent styles.

Table 2: Pairwise win-rates between four agents, with 95% confidence intervals. We begin with a purely supervised agent and then incrementally augment its training by adding self-play, reward shaping, and population training. Win-rates are from the perspective of a row agent and are estimated by conducting 2 000 games between each pair.

| | Behavior Cloning | + Self-Play | + Reward Shaping | + Population |
|---|---|---|---|---|
| **Behavior Cloning** | — | $31.2 \pm 2.0\%$ | $26.4 \pm 1.9\%$ | $26.4 \pm 1.9\%$ |
| **+ Self-Play** | $68.8 \pm 2.0\%$ | — | $28.1 \pm 2.1\%$ | $21.6 \pm 1.8\%$ |
| **+ Reward Shaping** | $\mathbf{73.6 \pm 1.9}\%$ | $71.9 \pm 2.1\%$ | — | $29.8 \pm 2.0\%$ |
| **+ Population** | $\mathbf{73.6 \pm 1.9}\%$ | $\mathbf{78.4 \pm 1.8}\%$ | $\mathbf{70.2 \pm 2.0}\%$ | — |

## 6.1 Emergent Behaviors

We empirically evaluate the emergent behaviors acquired by the agent throughout the self-play. The agent exhibits optimal expansion patterns in the early game, well-timed attacks, effective defense, and a strong instinct for locating the enemy base. Here, we showcase representative examples of these behaviors.

**Feints and Sidesteps.** When two armies are moving towards each other, or when one player chases the other, it is often beneficial to do "sidesteps" or other unexpected movements in order to confuse and outmaneuver opponents. Our agent shows exceptional maneuvering skill, with movements not seen even by the top players. These movements can be seen in the replays `[GR:1]`, `[GR:2]`, `[GR:3]`, `[GR:4]`.

**Snowballing.** Snowballing is a technique of incrementally converting a small lead into an even larger one. Our agent masters snowballing by timely attacks into the enemy land (converting enemy land into its own just before land-based army increment ticks), picking favorable trades, effective expansion of its borders, and unlike most players, our agent perceives the whole grid, making "intermezzo" moves that are away from the action just to make use of those investments later in the game. These patterns can be seen in the replays `[GR:5]`, `[GR:6]`, `[GR:7]`.

**Backdooring.** Our agent often creates "islands" of cells within enemy territory, from which it launches surprising attacks. Two of them can be seen in the replays `[GR:8]`, `[GR:9]`.

**Failure Modes.** In very few games, our agent exhibits suboptimal behavior, such as being stuck in dead ends formed by mountains and not knowing how to get out (`[GR:10]`, `[GR:11]`). We think that this happens because of the lack of look-ahead and the inability of the convolutional network to calculate distances, making it think that it can pass through some parts of the map while it cannot. Another failure mode is that the agent can get stuck in some specific "mode", for example the agent is focused only on either attacking, defense, or castle taking without balancing all three aspects. An example can be seen in the replay `[GR:12]`, where the agent starts capturing castles without realizing that it is being attacked.

## 7   Conclusion

We have presented a novel RTS research benchmark built on Generals.io that strikes a practical balance between accessibility and strategic richness. By supplying a customizable environment and a state-of-the-art reference agent, we open a new platform for algorithmic innovations and lower the barrier to entry for the DRL community. The analysis of emerging behaviors highlights both the strategic depth our environment provides, and the limitations of network architecture and learning approaches highlight possible directions for creating even stronger agents.

**Future Work.** We aim to extend the benchmark to include multi-team and free-for-all modes, thereby facilitating research into coordination, communication, and general-sum dynamics. To support these extensions and further scaling, we plan to adopt the `JAX` framework to enable an accelerator-agnostic, high-performance runtime. On the agent front, we plan to investigate policies parameterized by graph neural networks, which offer a more appropriate inductive bias given the game's inherent graphical structure. We also intend to employ game-theoretic approaches to reduce exploitability.

### Acknowledgements

This research was supported by the Charles University Grant Agency (GAUK), project no. 326525. Computational resources were provided by the e-INFRA CZ project (ID:90254), supported by the Ministry of Education, Youth and Sports of the Czech Republic.

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
