# OpenReview forum: "Artificial Generals Intelligence: Mastering Generals.io with Reinforcement Learning"
_rl-conference.cc/RLC/2025/Workshop/RLVG — RLVG Workshop - RLC 2025_

### Official Review · Reviewer_dJhy · 2025-06-13
**Nice benchmark, algorithm and paper with a somewhat over-promising title**

**Rating:** 4
**Confidence:** 4

**Summary:**

This paper proposes a new scientific benchmark based on the game Generals.io. The paper also suggests an RL based agent that it state-of-the-art compared to other algorithms and also plays at the same levels as expert humans. The paper is easy to read and follow and the examples of behaviours (like snowballing, feints, etc.) are very interesting. I think this is a nice contribution to the RL and games community and should be accepted into the workshop.

**Strengths:**

Strength:
- The benchmark is highly interesting to the community
- The algorithm that is SOTA on this benchmark

**Weaknesses:**

- The title suggest an AGI but there's not evidence of that

**Best Paper Nomination:**

No

**Claims:**

Claims:
- A new agent benchmark
- A state of the art RL algorithm

I would say yes to both

**Suggestions:**

- I would suggest to remove the mention of General Artificial Intelligence from the paper as I see no evidence to back this claim
- Link to the benchmark (I didn't find it in the paper)

---

### Official Review · Reviewer_ToUs · 2025-06-15
**Review for Generals.io**

**Rating:** 3
**Confidence:** 4

**Summary:**

The authors present an implementation of a popular "Generals.io" online game. They also present a PPO-like agent that can reach top-tier human performance on one of the game modes of the environment. The authors detail the training strategies and reward-shaping strategies used in the training of their agent. Benchmark results show that their agent outperforms several other methods in the environment.

**Strengths:**

The authors benchmark their method versus several other methods used previously in the game. These results show that their method tends to outperform previous methods.

**Weaknesses:**

While I think the environment is interesting from a game-playing perspective, I wonder if Generals.io actually provides challenges that are not already covered by existing environments.

**Best Paper Nomination:**

No

**Claims:**

The authors claim to provide an implementation of the popular Generals.io game as well as a learning agent that plays the game well. Both of these claimed contributions are well supported.

**Suggestions:**

I would encourage the authors to spend some time thinking about the challenges their environment provides that are not already covered by other existing environments.

---

### Decision · Program_Chairs · 2025-06-19

**Decision:**

Accept

**Comment:**

This paper introduces an implementation of the online game Generals.io as a new scientific benchmark, along with a PPO-like reinforcement learning agent that achieves top-tier human performance in one of the game modes.

Reviewers found this to be an interesting benchmark, mentioned the impressive performance of the proposed state-of-the-art RL algorithm against existing methods, and praised the paper's clear, easy-to-read presentation with engaging examples of learned behaviors.

However, the title was considered to be potentially over-promising, suggesting that solving the Generals.io benchmark would require general intelligence for which there is little evidence. If the title is not changed, it would help to better isolate the challenges that this benchmark poses compared to existing benchmarks in the abstract, introduction and conclusion of the paper and make the link to the benchmark more prominent. We encourage the authors to address this in the camera-ready version for presentation at the workshop.